# The Practices of Anaesthesiologists in the Management of Patients with Sickle Cell Disease: Empirical Evidence from Cameroon

**DOI:** 10.3390/healthcare9121617

**Published:** 2021-11-23

**Authors:** Dominique Djomo Tamchom, Aristide Kuitchet, Raymond Ndikontar, Serge Nga Nomo, Hermine Fouda, Luc Van Obbergh

**Affiliations:** 1Department of Anaesthesiology and Intensive Care, Douala Gynaeco-Obstetric and Paediatric Hospital, Douala 7072, Cameroon; 2Department of Anaesthesiology, Erasme Hospital, Free University of Brussels, 1070 Brussels, Belgium; luc.van.obbergh@erasme.ulb.ac.be; 3Faculty of Health Sciences, University of Buea, Buea 63, Cameroon; 4Faculty of Medicine, Free University of Brussels, 1070 Brussels, Belgium; 5Faculty of Medicine and Biomedical Sciences, University of Ngaoundéré, Ngaoundéré 454, Cameroon; fmsb@univ-ndere.com; 6Department of Anaesthesiology and Intensive Care, Yaoundé Gynaeco-Obstetric and Paediatric Hospital, Yaoundé 4362, Cameroon; raymond.ndikontar@fmsb.uy1.cm; 7Faculty of Medicine and Biomedical Sciences, University of Yaoundé I, Yaoundé 1364, Cameroon; hermine.ebana@fmsb-uy1.cm; 8Essos Hospital Centre, Department of Anaesthesiology and Intensive Care, Yaoundé 441, Cameroon; serge.nganomo@cnps.cm; 9Higher Institute of Medical Technology, Yaoundé 188, Cameroon

**Keywords:** sickle cell disease, anaesthesiologists, perioperative management, practices

## Abstract

Patients with sickle cell disease are more likely to undergo surgery during their lifetime, especially given the numerous complications they may develop. There is a paucity of data concerning the management of patients with sickle cell disease by anaesthesiologists, especially in Africa. This study aimed to describe the practices of anaesthesiologists in Cameroon concerning the perioperative management of patients with sickle cell disease. A cross-sectional study was carried out over four months and involved 35 out 47 anaesthesiologists working in hospitals across the country, who were invited to fill a data collection form after giving their informed consent. The data were analysed using descriptive statistics and a binary logistic regression model. Among the 35 anaesthesiologists included in the study, most (29 (82.9%)) had managed patients with sickle cell disease for both emergency and elective surgical procedures. Most of them had never asked for a haematology consultation before surgery. Most participants (26 (74.3%)) admitted to having carried out simple blood transfusions, while 4 (11.4%) carried out exchange transfusions. The haemoglobin thresholds for transfusion varied from one practitioner to another, between <6 g/dL and <9 g/dL. Only 6 (17.1%) anaesthesiologists had a treatment guideline for the management of patients with sickle cell disease in the hospitals where they practiced. Only 9 (25.7%) prescribed a search for irregular agglutinins. The percentage of haemoglobin S before surgery was always available for 5 (14.3%) of the participants. The coefficient (0.06) of the occurrence of a haematology consultation before surgery had a significant influence on the probability of management of post-operative complications (coefficient 0.06, 10% level of probability). This study highlights the fact that practices in the perioperative management of patients with sickle cell disease in Cameroon vary greatly from one anaesthesiologist to another. We disclosed major differences in the current recommendations, which support the fact that even in Sub-Saharan countries, guidelines applicable to the local settings should be published.

## 1. Introduction

Sickle cell disease is an autosomal recessive genetic disease resulting in an abnormal structure and quality of haemoglobin [1]. It is the most common genetic disorder, with over 50 million cases worldwide, and more than half of these cases (30 million) occur in Sub-Saharan Africa [2]. Most of these patients will undergo surgery during their lifetime [3] owing to a debilitating systemic syndrome characterised by chronic haemolytic anaemia, vaso-occlusive phenomena, ischaemia-reperfusion episodes with organ infarction, susceptibility to bacterial infection and chronic organ damage. Anaesthesiologists, therefore, play an important role in the perioperative management of patients with sickle cell disease, considering that many of them are operated upon and can develop postoperative complications. The main complications (febrile episodes, haemolysis, vaso-occlusive crises, thromboembolic situations and acute thoracic syndrome) are triggered by hypoxia, hypothermia, acidosis and pain. The surgical intervention in itself, as well as anaesthesia and analgesic procedures, are a challenge concerning the canon of management of patients with sickle cell disease, notably: rehydration, oxygenation, temperature regulation, and acid-base equilibrium [4]. In Senegal and Niger, significant complications have been noted in operated patients with sickle cell disease [5,6]. Up to 19% of operated patients develop vaso-occlusive complications [7,8]. These complications occur as a result of the surgical intervention itself or the underlying haematological disorders [3]. The rate of occurrence of complications can be reduced by optimizing patients’ conditions preoperatively and paying particular attention to patients’ situations during and after surgery [3,9]. Many studies have been carried out involving practices of health professionals on different aspects of management of patients with sickle cell disease, but little attention has been paid to anaesthesiologists in particular [10,11]. Considering the paucity of African data, our study aimed to describe the practices of anaesthesiologists in the management of patients with sickle cell disease in Cameroon.

## 2. Methods

### 2.1. Design

Our target population was anaesthesiologists practicing in Cameroon. Following the health pyramid in Cameroon, health facilities are organised into 3 levels: central, intermediate, and peripheral. The peripheral level includes district hospitals and health centres. The intermediate level includes regional hospitals. The central level includes reference centres, from highest (1st category) to lowest (3rd category). Most anaesthesiologists work in central-level health facilities and some in intermediate-level facilities. These are the health facilities that have the capacity, given their technical platforms, to receive patients with sickle cell disease for perioperative management. After approval from the Institutional Ethics Committee of Research for Human Health of the Gynaecologic-Obstetric and Pediatric Hospital of Douala (N°2019/0010/HGOPED/DG/CEI) and, the National Committee for Ethics in Research for Human Health (CNERSH) N° 2019/10/1196/CE/CNERSH/SP (Yaoundé, Cameroon), we performed a headcount of all anaesthesiologists in the national territory for four months running from 1 March 2019 to 30 June 2019. Criteria for the selection of participants included: being both a medical doctor and anaesthetist, as well as working in a health facility in Cameroon. Those who were not in the country during the study period were excluded (*n* = 12). We designed a data collection form to find frequent practices in the anaesthetic management of patients with sickle cell disease, which was submitted to and filled individually by participants. On this form, questions asked included the types of interventions carried out, the preparation for these interventions, details on blood transfusions, multidisciplinary management, the occurrence and the nature of complications and the need for ICU hospitalization after surgery. Discussions were held with participants when they filled their forms, or later on clear any doubts. All forms were filled after participants gave their informed consent. The data collection form is presented in Appendix A. We had no conflict of interest.

### 2.2. Statistical Analysis

Data were entered into a Microsoft EXCEL datasheet and analysed using the software IBM SPSS version 23. The binary logistic regression model was used to research probable factors influencing the practices of anaesthesiologists. Binary logistic regression best fits a non-linear relationship between variables and its probability lies between 0 and 1. For this study, this model hypothesised the extent to which the practices of anaesthesiologists (preoperative blood transfusion, occurrence of postoperative complications, and need for ICU hospitalisation after surgery) are influenced by factors such as gender; age; hospital category; haematology consultation before surgery; number of years in the practice of anaesthesia; and number of patients with sickle cell disease managed per year. The binary logistic regression model was therefore specified as:Z = α + β1Gender + β2Age + β3Hospital Category + β4 Haematology consultation before surgery + β5 Number of years in practice of anaesthesia + β6 Number of patients with sickle cell disease managed per year + μ.Z = Probability of preoperative blood transfusion or occurrence of post-operative complications or need for ICU hospitalization after surgery.β = Regression coefficient explaining changes caused in Z by changes in the independent variables.μ = Error term; α = Constant

## 3. Results

### 3.1. Characteristics of Participants

The characteristics of participants are shown in Table 1.

In total, 47 anaesthesiologists were identified in Cameroon, and 35 were included in our study. Out of these, 8.6% practiced in intermediate-level health facilities, 31.4% in category 1 reference hospitals, 37.1% in category 2 reference hospitals, and 22.9% in third category reference hospitals. The mean age of participants was 44 years, the range was from 35 to 62 years old. They were mostly men (54.3%). The average number of years in practice was 10 years (range: 2 to 34 years). The average number of patients with sickle cell disease managed by participant was 9 per year, with extremes varying from 1 to 40 per year.

### 3.2. Preoperative Management

Regarding the surgical setting and preparation of patients with sickle cell disease for anaesthesia (Table 2), 17.1% of anaesthesiologists had managed patients with sickle cell disease only for elective procedures while 82.9% had managed patients for both elective and emergency interventions. For elective interventions, 57.2% hospitalised their patients 24 h before surgery, 25.7% hospitalised 48 h before surgery, 11.4% hospitalised 72 h before surgery, and 2.7% hospitalised patients more than 72 h before surgery. Further, 22.8% of practitioners encountered only abdominal surgery, 74.3% encountered both abdominal or orthopaedic surgery, while 2.9% encountered only urological surgery.

In their daily practice, all the participants asked for a basic screening which includes: a complete blood count, ABO/Rhesus Blood group test, and a coagulation profile (aPPT). Only 25.7% prescribed a search for irregular agglutinins. The percentage of haemoglobin S before surgery was always available for 14.3% of the participants, 37.1% only sometimes, and 48.6% had never this result.

Only 22.9% of participants asked for a further haematology consultation before surgery.

### 3.3. Transfusion Practices and Postoperative Complications Encountered

In our study population, 74.3% of participants prescribed blood transfusions for the patients with sickle cell disease before surgery. Amongst those performing routine preoperative blood transfusion, 11.5% did not refer to any threshold and relied on other criteria such as surgery type and the clinical condition of the patient. The others used the thresholds varying from 6 to 9 g/dL (see Table 3 for further details). Only 11.4% (4/35) performed exchange transfusion before surgery.

Almost half of them (45.7%) registered postoperative complications. Amongst these, some had noted only one type of complication: 25% noted vaso-occlusive crises, 12.5% acute chest pain syndrome, 18.7% poorly tolerated anaemia, 12.5% retarded awakening from anesthesia, 6.2% pulmonary embolism, and 6.25% postoperative infection. The rest had noted at least two types of complications (see Table 3). Likewise, 51.4% used postoperative ICU management for their patients, while 8.5% reported deaths.

Only 17.1% had management guidelines for patients with sickle cell disease in the hospitals where they practiced.

### 3.4. Factors Influencing Practice

The factors influencing the practices of anaesthesiologists are presented in (Table 4). The results show that the number of patients with sickle cell disease managed per year (*p* value = 0.06) significantly influenced the probability of preoperative blood transfusion at a 10% level of probability. Haematology consultation before surgery (*p* value = 0.09) and male gender (*p* value = 0.06) significantly influenced the probability of management of postoperative complications at a 10% level of probability, respectively. Further, the number of years in the practice of anaesthesia (*p* value = 0.09) and the number of patients with sickle cell disease managed per year (*p* value = 0.04) significantly influenced the probability of needing ICU hospitalisation after surgery at the 10% and 5% level of probability, respectively. The more experienced anaesthesiologists tended to request an ICU bed compared with the less experienced.

The rules of thumb of the binary logistic regression for this study state that significant variables have an influence on the preoperative blood transfusion, occurrence of postoperative complications as noted by anaesthesiologists, and the likelihood of an anaesthetist ordering an intensive care unit admission after surgery. The *p* values of <0.05 and <0.1 represented statistically significant dependence at the 5% level and 10%, respectively.

## 4. Discussion

Limited studies have shown the practices of anaesthesiologists in the management of patients with sickle cell disease in Sub-Saharan Africa. These practices varied from one anaesthesiologist to another, despite almost similar constraints related to the environment.

Most anaesthesiologists had managed emergency cases involving patients with sickle cell disease. In emergencies, preoperative risks are higher in patients with sickle cell disease. The urgent nature of the surgery, combined with other factors, leads to a higher incidence of red blood cell sickling [12]. Cultural practices, such as traditional treatments and other phytotherapy measures, are common and constitute a major obstacle to early medical management. This results in a delay in seeking medical care, explaining the tendency for emergency surgery in patients with sickle cell disease observed in this study.

Most participants had treated patients with sickle cell disease for abdominal and orthopaedic surgery. This result was consistent with data in the literature [3,13,14,15] which classify hepatobiliary, digestive, and bone complications as the most common surgical indications for patients with sickle cell disease.

The patients were hospitalised in most cases, 24 h before elective surgery. Patients with sickle cell disease should be admitted to the hospital for elective surgery several days before the procedure to ensure better preparation and thus limit the occurrence of complications. The clinical status of these patients may require lengthy preparation. Many authors recommend admission of patients with sickle cell disease at least one day prior the surgery, to allow time for preoperative work-up and intravenous hydration [16,17,18,19,20].

Preoperative check-ups were limited to a complete blood count, coagulation profile, and blood grouping/rhesus factor. Several factors may contribute to this, such as the urgency of the surgery and the availability of technical facilities. The low proportion of testing for irregular agglutinins could be explained by the costs of these tests, which are, most of the time, paid by the patients and their families. Other causes could be the inadequate technical facilities in the laboratories of most hospitals where the anaesthesiologists work, as well as an emergency that would not allow enough time for more tests. This blood test should be is strongly recommended for the management of patients suspected to have delayed post-transfusion haemolysis [21]. Thus, given the risk of alloimmunization of red blood cells caused by non-standardised blood bank systems and the transfusion reactions observed in several regions of sub-Saharan Africa [22,23], all patients with sickle cell disease require a full blood count, urea and electrolytes and group and full red cell antibody screen before surgery [24].

The multidisciplinary care of patients with sickle cell disease is a major asset in the improvement of their health. The difficulty in this context is the access to specialist doctors in all regions. The number of specialist doctors (including haematologists) is small and only a few first-category hospitals have them in various fields. This could explain why most anaesthesiologists are the only doctors involved in the perioperative management of patients with sickle cell disease. The non-emergency context of surgery, and optimal multidisciplinary perioperative management, associating anaesthesiologists, surgeons and haematologists, contributed in part to prevent the occurrence of postoperative complications linked to sickle cell disease in several studies [6,15]. This collaboration makes it possible to assess the severity and functional impact of the disease in patients with sickle cell disease, firstly according to the vaso-occlusive and transfusion history and secondly according to the degree of progression of degenerative complications [25].

Preoperative management of patients with sickle cell disease frequently includes red blood cell transfusion to reduce the risk of morbidity and mortality associated with the operation [26]. However, there is currently no consensus on the benefit of preoperative transfusion in non-symptomatic patients [27]. The majority of anaesthesiologists in our series were accustomed to preoperative blood transfusion, and few used exchange transfusion. The benefit of transfusion in patients with sickle cell disease is the rapid reduction of the proportion of red blood cells containing haemoglobin S, which stops the noxious pathophysiological cascade. However, it entails particular risks in these patients: paradoxical worsening of the clinical state by hyperviscosity when the haemoglobin rises to a very high value (it should never exceed 10–11 g/dL); frequent alloimmunization due to the constitutional differences in blood group antigens [28]; and iron overload. These situations could justify the attitude of those anaesthesiologists in our study, who were not used to preoperative blood transfusion in patients with sickle cell disease.

Blood transfusion is rarely proposed to increase haemoglobin levels, as anaemia in patients with sickle cell disease is chronic and most often well-tolerated. In our study, this could justify the variability of the transfusion threshold from one anaesthesiologist to another. During a prospective cohort study of 1516 patients with sickle cell disease in a Tanzanian hospital, one of the independent risk factors leading to death was low haemoglobin <5 g/dL [29]. As observed in our study, criteria such as surgery type and the clinical condition of the patient may be decisive in choosing the transfusion threshold in the preoperative period.

Postoperative complications of sickle cell disease are highly challenging situations and their frequency depends on the condition of the patients before surgery, the type of surgery performed, and the teams involved [25]. The most observed are febrile episodes, haemolysis, vaso-occlusive crises, thromboembolic situations, acute chest pain syndrome [24]. Postoperative care for patients with sickle cell disease requires appropriate resuscitation to avoid the occurrence of all factors that can trigger sickle cell disease of red blood cells and the occurrence of complications. In a study performed in Niger [6], all patients were admitted to the intensive care unit for continuous monitoring during the first 24 h or more. A retrospective review of all patients with sickle cell disease who had splenectomy between 1999 and 2007 in Saudi Arabia [30] showed that there was no real benefit from routine perioperative admission to the ICU. Using the ICU in postoperative care for these patients could depend on the team, but also the context.

The wide clinical spectrum of sickle cell disease implies that despite exit, personalised management should always be allowed [31]. Although the quality of recent perioperative clinical studies has improved, many questions concerning ideal management remain unanswered [31]. Nevertheless, guidelines aim to harmonise and improve practices based on well-established data. This enables the teams to register the events and to readjust their practice according to the results observed.

Some factors specific to each anaesthesiologist may influence practices as observed in our study. The preoperative haematology consultation reduces the likelihood of a complication occurring postoperative. The male anaesthesiologists have a higher probability of observing postoperative complications; although this was outlined by our results, we did not found an explanation for this. The specialised approach of haematologists in the preparation of these patients would reduce the occurrence of postoperative complications.

The higher the number of patients per year and years of anaesthesia practice, the greater the likelihood of use of postoperative intensive care. This could be explained by the attitude of most experienced anaesthesiologists who use postoperative intensive care immediately after the procedure, to anticipate well-known predictable complications in patients with sickle cell disease, but also by the vulnerability of patients with sickle cell disease, who in the postoperative period may often present complications not only related to sickle cell disease but also surgery.

Our study had some limitations. First, the retrospective nature of the study and, secondly the fact that it was only performed in one country; bearing this in mind, a more widespread study including multinational caregivers will be useful to confirm our results.

## 5. Conclusions

The practices of anaesthesiologists in the perioperative management of patients with sickle cell disease in Cameroon vary greatly and are significantly influenced by factors such as the number of patients treated per year, haematological consultation before surgery, and the number of years in practice. The attitudes of the most experienced physicians in the profession tend to differ from other physicianss. Emergency surgeries in patients with sickle cell disease are quite frequent and the most common surgical procedures are abdominal and orthopaedic. The preoperative check-up is most often limited to a complete blood count, ABO/Rhesus blood group test, and a coagulation profile. Preoperative transfusions are frequent, but exchange transfusions are rare. The thresholds and criteria for transfusions vary greatly; this reflects the fact that only a minority of the responders have a management protocol for patients with sickle cell disease. Teaching, education and guidelines with audits and follow-up are highly needed. It would therefore be beneficial to set up guidelines that are easily applicable in this specific context; the involvement of haematologists and sickle cell specialists in the process to this end would certainly be a useful addition.

## Figures and Tables

**Table 1 healthcare-09-01617-t001:** Socio-demographic characteristics of anaesthesiologists.

Variables	Number (*n* = 35)	Percentage (%)	Mean	Min	Max	Standard Deviation
**Gender**						
Male	19	54.3				
Female	16	45.7				
**Age**			44	35	62	8.310
**Number of years in the practice of anaesthesia**			10	2	34	8.434
**Number of patients with sickle cell disease managed per year**			9	1	40	7.629
**Hospital Category**						
- Intermediate level	3	8.6				
- Central level-first category	11	31.4				
- Central level-second category	13	37.1				
- Central level-third category	8	22.9				

**Table 2 healthcare-09-01617-t002:** Preoperative management of patients with sickle cell disease by anaesthesiologists.

Variables	Number (*n* = 35)	Percentage (%)
**Context of surgery**		
Elective surgery only	6	17.1
Both emergency and elective surgery	29	82.9
**Time interval between admission and surgery (elective)**		
Admission 24 h before surgery	20	57.2
Admission 48 h before surgery	9	25.7
Admission 72 h before surgery	4	11.4
Admission for more than 72 h before surgery	2	5.7
**Types of surgical interventions encountered**		
Abdominal surgery	8	22.8
Abdominal and Orthopedic surgery	26	74.3
Urology only	1	2.9
**Preoperative workup**		
Full blood count, ABO/Rhesus blood group, and coagulation profile (aPPT).	35	100
**Requested a search for irregular agglutinins**		
Yes	9	25.7
No	26	74.3
**Preoperative knowledge of the percentage of HbS**		
Always	5	14.3
Sometimes	13	37.1
Never	17	48.6
**Haematology consultation before surgery**		
Yes	8	22.9
No	27	77.1

**Table 3 healthcare-09-01617-t003:** Perioperative blood transfusion, postoperative complications encountered and need for ICU hospitalisation.

Variables	Anaesthesiologists (*n* = 35)	Percentage (%)
**Preoperative blood transfusion**		
Yes	26	74.3
No	9	25.7
**Threshold for preoperative transfusion (*n* = 26)**		
Hb < 6 g/dL	2	7.7
Hb < 7 g/dL	16	61.5
Hb < 8 g/dL	4	15.4
Hb < 9 g/dL	1	3.9
Clinical criteria	3	11.5
**Use of preoperative exchange transfusion (*n* = 35)**		
Yes	4	11.4
No	31	88.6
**Occurrence of post-operative complications (*n* = 35)**		
Yes	16	47.5
No	19	52.5
**Complications encountered by anaesthesiologist (*n* = 16)**		
Vaso-occlusive crisis	4	25
Acute chest syndrome	2	12.5
Pulmonary embolism	1	6.2
Poorly tolerated anaemia	3	18.7
Postoperative infection	1	6.2
Postoperative vaso-occlusive crisis and acute chest pain syndrome	1	6.2
Poorly tolerated anaemia and infection	1	6.2
Retarded awakening and poorly tolerated anaemia	2	12.5
Postoperative vaso-occlusive crisis and infection	1	6.2
**Need for ICU hospitalisation** **after surgery (*n* = 35)**		
Yes	18	51.4
No	17	48.6
**Availability of hospital guidelines for the management of patients with sickle cell disease (*n* = 35)**		
Yes	6	17.1
No	29	82.9

**Table 4 healthcare-09-01617-t004:** Factors influencing preoperative blood transfusion, management of postoperative complications, and need for ICU hospitalisation after surgery.

	Preoperative Blood Transfusion			Management of Postoperative Complications Noted by Anaesthesiologists			Likelihood of an Anaesthetist Ordering an Intensive Care Unit Hospitalisation.		
Variables	Coefficient	Wald	*p* Value	Coefficient	Wald	*p* Value	Coefficient	Wald	*p* Value
-Male (*n*)	2.4	2.2	0.1	2.3 *	3.5	0.06	1.2	1.03	0.3
-Age (years)	−0.05	0.04	0.8	−0.2	2.1	0.1	−0.3	2.8	0.09
-Hospital Category									
first category central	−19.8	0.0	0.9	−21.1	0.0	0.9	−23.1	0.0	0.9
second category central	−16.7	0.0	0.9	−24.2	0.0	0.9	−23.1	0.0	0.9
third category central	−20.1	0.0	0.9	−23.4	0.0	0.9	−44.9	0.0	0.9
-Haematology consultation before surgery (*n*)	−1.2	1.8	0.4	−2.7 *	2.8	0.09	−20.2	0.0	0.9
-Number of years in the practice of anaesthesia (*n*)	0.2	0.4	0.5	0.1	1.1	0.2	0.3 *	2.7	0.09
-Number of patients managed per year (*n*)	0.4 *	3.3	0.06	0.08	0.7	0.3	0.3 **	4.2	0.04
Constant	18.1	0.00	0.9	32.3	0.0	0.9			
χ^2^	17.2 **			18.7 **			22.7 **		
Classification table	85.7			88.6			82.9		
Cox & Snell R square	0.3			0.4			0.6		
Nagelkerke R squared	0.5			0.5			0.4		
-2 Log likelihood	20.4			29.0			25.4		

N.B ** and * are significant at 5% and 10%, respectively.

## Data Availability

The data presented in this study are available on request from the corresponding author.

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
