# Peer review of "The Practices of Anaesthesiologists in the Management of Patients with Sickle Cell Disease: Empirical Evidence from Cameroon"

_healthcare, 2021, doi:10.3390/healthcare9121617_

Round 1

Reviewer 1 Report

Dear authors,

thank you for performing the present investigation. 

Here you present results of a survey done on anesthesiologists who are treating sickle cell diseased patients. 

The topic is interesting, but the manuscript is at some points confuse and may be read misleading, because it is not always clear that you are presenting numbers of anesthesiologists and not on patients. 

In the discussion you could improve by more concrete examples and explanations. 

Please find more comments in the attached pdf. 

Kindly

Author Response

Response to Reviewer 1 Comments

Point 1 - Title: please avoid this term, (Sicklers) because it could hurt people. Choose some more professional word.

Response 1: We replace this term by (patients with sickle cell disease).

“The Practices of Anaesthesiologists in the management of patients with sickle cell disease: Empirical Evidence from Cameroon”.

Point 2 - Line 24, Line 26: re-word: asked or invited to...,

Response 2: Has been reworded (invited to…)

Has been replace (patients with sickle cell disease)

Point 3 - Line 46, Line 47: maybe add here the reason: (ischemic events due to sickle cell disease...). “Most of these patients will undergo surgery during their lifetime”….

Response 3: Has been added

Most of these patients will undergo surgery during their lifetime “due to a debilitating systemic syndrome characterised by chronic haemolytic anaemia, vaso-occlusive phenomena, ischaemia-reperfusion episodes with organ infarction, susceptibility to bacterial infection and chronic organ damage”.

Point 4 – Lines 47 to 49 : better: perioperative because anaesthesiologists do not influence surgery...

Again, be more concrete. give one or two sentences on the complications and their triggers.

Response 4:  Anaesthesiologists, therefore, play an important role in the “perioperative” management of sicklers. The main complications (febrile episodes, haemolysis, vaso-occlusive crises, thromboembolic situations and acute thoracic syndrome) are favoured by hypoxia, hypothermia and pain

Point 5 Line 52 to 56 : this is all true, but seems a little confuse. Please put it in the right time line.

explain the frequency and the occurrence of complications, what kind of complications and how are they triggered. then deal with prevention and/or treatment.

Response 5: In Senegal and Niger, significant complications have been noted in operated patients with sickle cell disease [8,9]. Up to 19% of operated patients develop vasocclusive complications [4,6]. These complications occur as a result of the surgical intervention itself, or underlying haematological disorders [3]. The rate of occurrence of complications can be reduced by optimizing patients' conditions preoperatively and paying particular attention to patients' situations during and after surgery [3,7].

Point 6 Line 64 to 68 : “Following the Health pyramid in Cameroon, health facilities are organized into 3 levels: central, intermediate, and peripheral. The peripheral level includes district hospitals and health centres. The intermediate level includes regional hospitals. The central level includes reference centres, from highest (1st category) to lowest (3rd category)”.

why do you mention this? Maybe to explain to exposure of the anaesthesiologists to sickle cell disease? If so then give the implication:

where do you find anesthesia and surgery?

Response 6 : The majority of anaesthesiologists works in central level health facilities and someone in intermediate level facilities. These are the health facilities that have the capacity, given their technical platforms, to receive patients with sickle cell disease for perioperative management.

Point 7 Line 69 : After approval from our institutional ethical committees for research”…

do you have more than one?

Please give IRB number.

Response 7 : we have two ethical committees.

After approval from the Institutional Ethics Committee of Research for Human Health of the Gynaecologic-Obstetric and Pediatric Hospital of Douala (N°2019/0010/HGOPED/DG/CEI) and, the National Committee for Ethics in Research for Human Health (CNERSH) N° 2019/10/1196/CE/CNERSH/SP.

Point 8 Line 72 : better: We designed a data collection...

Response 8 : Has been reworded.

“We designed a data collection form to find frequent”…

Point 9 Line 77 : I guess also the nature of the complications?

Response 9 : Yes. Has been added

On this form, questions asked included the types of interventions carried out, the preparation for these interventions, details on blood transfusions, multidisciplinary management, the “nature” and the occurrence of complications.

Point 10 - Figure 1. : did really all colleagues fill the form?

Response 10 : Yes, all the colleagues fill the form.

Point 11 - Line 91 : this should be mentioned in the data you collect : “need for ICU hospitalization after surgery”

Response 11 : Has been mentioned

On this form, questions asked included the types of interventions carried out, the preparation for these interventions, details on blood transfusions, multidisciplinary management, the nature and the occurrence of complications and the “need for ICU hospitalization after surgery”.

Point 12 - Line 108 : please make clear these are the anaesthesiologists and not patients.

Table 1. Socio-demographic characteristics of “participants”.

Response 12 : Precision had been maked.

Table 1. Socio-demographic characteristics of “anaesthesiologists”.

Point 13 - Line 119 : do you mean elective? If so please be consistent.

For “programmed” interventions, 57.2% hospitalized their patients…

Response 13 : Yes, we mean elective

For “elective” interventions, 57.2% hospitalized their patients…

Point 14 - Table 2 : Both emergency and “programmed” surgery

Response 14 :  Both emergency and “elective” surgery

Point 15 : any other workup done?

ECG, labs?

Response 15 : It was a question of the minimum assessment regularly requested in view of the context of limited resources of the patients who support their own care. We had not asked about the completion of the other workup.

Point 16  - Line 128: I guess you mean aPTT?

Blood group test, and a coagulation profile (Prothrombin time and activated cephalin time).

Response 16 : Yes, it is aPTT.

Blood group test, and a coagulation profile (aPTT).

Point 17  - Line 149 : “Need” “n”

Response 17 : “need”

Point 18  - Table 3, Page 5: Now it gets confusing. Do you mean the population as anesthetists or patients?  “Population” (n)

Response 18 : We mean the population as anaesthesiologists :Anaesthesiologists” (n)

Point 19  - Table 3, Page 6 : are these patients or anesthetists who faced these complications? This should be very clear, please. “Complications encountered (n = 16)”

Response 19 : It is the anaesthetists who have observed these complications during their practice. “Complications encountered by anaesthesiologists (n = 16)”

Point 20  - Lines 151 and 153 :  what is this number? (0.06)

Response 20 : 0.06 is p Value.

(P value = 0.06)

Point 21  - Table 4 : although this is very complete this table is confusing the reader. Please reduce the table to the meaningful numbers and explain how to interpret it.

Again, it's the anesthetists and not the patients data?

Response 21 : The rules of thumb of the Binary logistic regression for this study states that significant variables have an influence on the preoperative  blood transfusion, Occurrence of post-operative complications as noted by anaesthesiologists, and the  Need for ICU hospitalization after surgery (likelihood of an anaesthetist ordering an intensive care unit). A value of p < 0.05 and p < 0.1 representing  statistically significant dependence at the 5% level and 10% respectively.

Point 22  - Line 162 : please avoid this non-sense

 “To the best of our knowledge” , this is the first study showing the practices of anaesthesiologists in the management of patients with sickle cell disease in Sub-Saharan Africa.  

Response 22 : Has been replace by :

Limited studies showed the practices of anaesthesiologists in the management of patients with sickle cell disease in Sub-Saharan Africa.

Point 23  - Line 164 : this should be explained later in the text. what restrictions are you facing.

Response 23 : will be explained later in the text.

The main constraints to which anesthetists are subjected are the limitation of technical platforms for explorations and treatment, as well as the limited resources of most of the patients treated.

Point 25  - Line 169 - 170 : although understandable, it sounds double. Is the urgency now a contributor to surgery or a cause of surgery?

(This trend towards emergency surgery for sicklers in this study could be explained, on the “one hand by the complications of sickle cell disease”, and on the other hand by the fact that in our context, cultural practices such as traditional treatments and other phytotherapy measures are a major obstacle to early medical care, resulting in most patients being diagnosed late with emergency surgical indications).

Maybe just shuffle this sentence: first explain the common practice with late medical care as a result then mention the high urgency as a result of this?

Response 25 : We have reworded the sentence.

Cultural practices such as traditional treatments and other phytotherapy measures are common and constitute a major obstacle to early medical management. This results in delayed medical care, explaining the tendency for emergency surgery in patients with sickle cell disease in this study.

Point 26  - Line 181 : this is a quite old paper. is there nothing more recent to refer to?

“A review of a series of patients with sickle cell disease who underwent cholecystectomy from 1978 to 1991”…

Response 26 : Has been replaced by more recent papers

“Many authors recommend admission of patients with sickle cell disease at least one day prior the surgery, to allow time for preoperative work-up and intra-venous hydration[18–22]”

Point 27  - Line 188 : do you mean they have to pay extra for this? If so, write it.

(The low testing for irregular agglutinins could be explained by the costs of these investigations, “which most of the time are borne by the patients and their families”).

Response 27 :

The low testing for irregular agglutinins could be explained by the costs of these “tests”, which most of the time are “paid” by the patients and their families.

Point 28  - Line 209 : if the patients are non-symptomatic. If they have pain then the indication is clear...

(However, there is currently “no consensus on the benefit of preoperative transfusion in these patients”).

Response 28 :  We add the precision.

However, there is currently no consensus on the benefit of preoperative transfusion in “non-symptomatic patients”.

Point 29 - Line 217 : this implies another argument ...on the one hand...on the other hand...

please re-phrase

(These situations could, on the one hand, justify the attitude of those anaesthetists in our study, who had not been used to preoperative blood transfusion in sicklers).

Response 29 :  Has been rephrased

These situations could justify the attitude of those anaesthetists in our study, who had not been used to preoperative blood transfusion in patients with sickle cell disease.

Point 30 : What about the age and co-morbidities such as cardiovascular disease?

Response 30 : We did not address these aspects in this study focused on anaesthetists. We would like to do so in another study which will focus directly on patients with sickle cell disease.

Point 31 : you should refer to the very recently published guideline in "Anaesthesia" by Walker et al. 2021, 76,805-817.

This might be discussed even more for your context and restrictions.

Response 31 :  We have referred to this article to further support our discussion.

Point 32 – Line 227 : which? sickle cell?

Post-operative complications of sickle cell disease are highly feared situations and their frequency depends “on the disease”, the type of surgery performed, and the teams involved

Response 32 :  We wanted to talk about condition of the patient surgery. We have rephrased:

Post-operative complications of sickle cell disease are highly feared situations and their frequency depends “on the condition of the patients before surgery”, the type of surgery performed, and the teams involved.

Point 33 – Line 232, 233 : what kind of special resuscitation?

all patients were admitted to the intensive care unit “for special resuscitation” during the first 24 hours or more.

Response 33 :  We wanted to talk about continuous monitoring. We have rephrased :

all patients were admitted to the intensive care unit “for continuous monitoring” during the first 24 hours or more.

Point 34 – Line 241 : I guess you mean to register the events? please rephrase

This enables the “teams to take stock of the” difficulties en countered and to readjust according to the results observed implementation.

Response 34 :  Yes. We have rephrased:

This enables the “teams to register the events” and to readjust their practice according to the results observed. 

Point 35 – Line 244, 245, 246 : why do you put this is one sentence/paragraph? is there any explanation for this?  one could think the males are more stubborn and would not consult a haematologist???

“The preoperative haematology consultation reduces the likelihood of a complication occurring postoperative, while male anaesthesiologists have a higher probability of observing postoperative complications”.

Response 35 : Following the reviewer's remark, we scored a full stop after the first idea to get two sentences.

The preoperative haematology consultation reduces the likelihood of a complication occurring postoperative. The male anaesthesiologists have a higher probability of observing postoperative complications, although this was outlined by our results we did not found an explanations for this.

Point 36 – Line 244 : female”

Response 36 :  No more appropriated

Point 37 – Line 248, 249 : this is an assumption, which may be correct, but there is no evidence for. Maybe report it as assumption or possible explanation...

“Female anaesthesiologists would tend to be more attentive and to better observe preventive measures of postoperative complications in sicklers than their male counter parts”.

Response 37 : Has been delated.

Point 38 – Line 257, 258 : what is second?

what about a study on caregivers but not on patients?

First, the retrospective nature of the study and performed in only one country, bearing this in mind a more widespread study can be useful to confirm our results.

Response 38 : Has been rephrased.

First, the retrospective nature of the study and, “secondly” the fact that it has been performed in only one country; bearing this in mind, a more widespread study including multinational caregivers can be useful to confirm our results.

Point 39 – Line 270 : Maybe you could highlight this even more, please. “Teaching, education and guidelines with audits and follow-up is highly needed”.

The thresholds and criteria for transfusions vary greatly, this reflects the fact that only a minority of the responders have a management protocol for patients with sickle cell disease.

Response 39 : Has been highlighted as reviewer suggested.

The thresholds and criteria for transfusions vary greatly, this reflects the fact that only a minority of the responders have a management protocol for patients with sickle cell disease. Teaching, education and guidelines with audits and follow-up are highly needed.

Point 40 – Line 369 : “ow” (H)

Response 40 : Has been corrected. “How”

Point 41 – Line 374 : patient-age?

“The Age groups”

Response 41 : Has been corrected. “ patient-age”

Point 42 – Line 406 : opioid. Morphine is an opioid itself.

Response 42 : Has been corrected. “Opioid”

Point 43 – Line 406 : muscle relaxants?

Response 43 : Yes. Has been corrected. “muscle relaxants”

Reviewer 2 Report

Many thanks for having let me reviewed this article. I found it very interesting, and I think it could be of interest of the readers of Healthcare as the management of sicklers is very specifical, involves frequently anesthesiologists in the perioperative period, and is described here from a country where frequency of sicklers is relatively high.

I have minor comments only:

I would modify the Flowchart :

- can you detail why 12 anesthesiologists are not meeting inclusion criteria?

- in my opinion, this diagram is a bit detailed, as for a randomized controlled trial. I would simplify it.

Line 87: I would insert the paragraph “model specification” inside the paragraph “statistical analysis”

Line 107: remove the brackets

In the tables: in my opinion, the word “Frequency” is not appropriate, I would change it for “number”.

It would be interesting (even if we understand it is 37) to precise the total number of participants in the first line of each table (n=37).

Author Response

Response to Reviewer 2 Comments

Point 1 – Figure 1 : I would modify the Flowchart :

- can you detail why 12 anaesthesiologists are not meeting inclusion criteria?

- in my opinion, this diagram is a bit detailed, as for a randomized controlled trial. I would simplify it.

Response 1 : We agree with your comment although the editor did not allow us to modify this figure:

We therefore added in line 84 of the method: “Those who were not in the country during the study period were excluded (n=12).”

Point 2 – Line 87 : I would insert the paragraph “model specification” inside the paragraph “statistical analysis”

Response 2 : Has been done as suggested by reviewer.

Point 3 – Line 107 : remove the brackets

Response 3 : Has been remove

Point 4 : In the tables: in my opinion, the word “Frequency” is not appropriate, I would change it for “number”.

It would be interesting (even if we understand it is 35) to precise the total number of participants in the first line of each table (n=35).

Response 4 : Has been done as suggested by reviewer.

Reviewer 3 Report

  1. Binary logistic regression:

What’s the Sig. value for Omnibus tests of model coefficients? Does it < 0.0.5 or not? Authors need to present this information to see how this model fitting your data.

Did author try Hosmer and Lemeshow test? What’s the Sig. value for this test?

Which method did author use for variables in the equation? Enter, Forward or Backward?

Please describe the rationale to use 90% CI for Exp(B).

  1. The result showed in table 4, “need for ICU hospitalization after surgery” I don’t think it is a perfect description according to your questionary. It’s more like the probability of an anesthesiologist ordered for ICU hospitalization. Please consider to choose a more fitting title for this statement.

And also, author use a model hypothesized the extent to which the practices of anaesthesiologists (preoperative blood transfusion, Occurrence of postoperative complications, and need for ICU hospitalization after surgery) are influenced by factors such as Gender; Age; Hospital category; Haematology consultation before surgery; Number of years in the practice of anaesthesia; and Number of sicklers managed per year. The item “occurrence of postoperative complications” is not a practice of anesthesiologist.

  1. Line 152-154: “Haematology consultation before surgery (0.06) and the gender male (0.06) significantly influence the probability of occurrence of postoperative complications at a 10% level of probability respectively.” The Haematology consultation before surgery Sig. value is 0.09 according to the result showed in table 4.
  2. Line 191-192: Please check the sentence.
  3. Did author try to analyze the relationship between “Occurrence of post-operative complications” and “Haematology consultation before surgery”? Is there a significant difference? This is might needed for the future protocol drafting of care for sicklers.

6.Please check your citations and make sure they meet the requirement of Health Care.

Author Response

Response to Reviewer 3 Comments

Point 1 : Binary logistic regression

Reviewer Question: What is the sig value for omnibus tests of the model coefficient?

Answer : The significant values of the omnibus tests were (0.028 significant at 5%); (0.016 significant at 5%), and (0.004 significant at 1%) for preoperative blood transfusion, occurrence of postoperative complications and need for ICU hospitalization after surgery respectively.

Reviewer Question: did author try Hosmer and Lemeshow tests?

Answer : The values of Hosmer and Lemeshow tests were not significant for preoperative blood transfusion (significance value is 0.4), occurrence of postoperative complications (significance value is 0.5), and need for ICU hospitalization after surgery (significance value is 0.61) respectively. These indicate the goodness of fit of the model.

Also, the values of the classification table as shown in the result table indicating that on the average, the model for preoperative blood transfusion, occurrence of postoperative complications and need for ICU hospitalization after surgery respectively were (85.7 against 77.1); (88.6 against 57.1), and (82.9 against 54.3 percent). Hence, confirming the goodness of fit or appropriateness of the model selected.

Reviewer Question: which method did author use for variable equation? Enter, Forward or Backward?

Answer : The method enter was used for the variables equations because it allows all the variables to enter in a single step unlike the stepwise regression (backward or forward), at each step, the independent variable not in the equation that has the smallest probability F is entered, if that probability is sufficiently small. Also, variables already in the equation are removed if their probability F becomes sufficiently large and the method terminates when no more variables are eligible for inclusion or removal.

Reviewer Question: Please the rationale to use 90% for exp (B).

Answer : The level of the confidence interval was chosen by the research team following our sample size where we have low power to detect an effect, and also the confidence that the values of the results will fall between the upper and the lower limits if the procedure or research is repeated again, and also following Hair (2009) and Harzelrigg (2009). For instance, Hair (2009) states that establishing the significance level denotes the chance the researcher is willing to take of being wrong about whether the estimated coefficient is different from zero. Increasing the significance level to higher value (10%) allows for a larger chance, and also makes it easier to conclude the coefficient is different from zero. Harzelrigg (2009) states that when setting confidence intervals, there is nothing sacrosanct or magical about these numbers, either Z or alpha. They are entirely conventional choices, and one is free to select a different number depending on the nature of the study (1%, 5%, and 10%).

Point 2 : The result showed in table 4, “need for ICU hospitalization after surgery” I don’t think it is a perfect description according to your questionary. It’s more like the probability of an anesthesiologist ordered for ICU hospitalization. Please consider to choose a more fitting title for this statement.

And also, author use a model hypothesized the extent to which the practices of anaesthesiologists (preoperative blood transfusion, Occurrence of postoperative complications, and need for ICU hospitalization after surgery) are influenced by factors such as Gender; Age; Hospital category; Haematology consultation before surgery; Number of years in the practice of anaesthesia; and Number of sicklers managed per year. The item “occurrence of postoperative complications” is not a practice of anesthesiologist.

Response 2 : Has been changed as suggested by reviewer.

“likelihood of an anaesthetist ordering an intensive care unit hospitalization”.

We replace “occurrence of complications” by “management of postoperative complications”

Point 3 - Line 152-154 : “Haematology consultation before surgery (0.06) and the gender male (0.06) significantly influence the probability of occurrence of postoperative complications at a 10% level of probability respectively.” The Haematology consultation before surgery Sig. value is 0.09 according to the result showed in table 4.

Response 3 : Has been corrected according to the corresponding Sig. value in Table 4. “Haematology consultation before surgery (0.09)”

Point 4 - Line 191-192: Please check the sentence.

This blood test should be is strongly recommended for the management of patients suspected to have delayed post-transfusion haemolysis [19] given the risk of alloimmunization of red blood cells caused by non-standardized blood bank systems and the transfusion reactions observed in several regions of sub-Saharan Africa [20,21].

Response 4 : We added a full stop after [19].

This blood test should be is strongly recommended for the management of patients suspected to have delayed post-transfusion haemolysis [19]. This, given the risk of alloimmunization of red blood cells caused by non-standardized blood bank systems and the transfusion reactions observed in several regions of sub-Saharan Africa [20,21].

Point 5 : Did author try to analyse the relationship between “Occurrence of post-operative complications” and “Haematology consultation before surgery”? Is there a significant difference? This is might needed for the future protocol drafting of care for sicklers.

Response 5 : Yes we did :

Haematology consultation before surgery (P value = 0.09) significantly influence the probability of “management” or occurrence of postoperative complications at a 10% level of probability.

Point 6 : Please check your citations and make sure they meet the requirement of Health Care.

Response 6 : Has been checked

Round 2

Reviewer 1 Report

Dear authors,

thank you for adopting the suggestions. 

The manuscript became more readable and clear by this.

I have only some minor points added in the pdf. 

Best wishes

Reviewer 3 Report

Authors used binary logistic regression method to predict the practices of anesthesiologists behavior (preoperative blood transfusion, Occurrence of postoperative complications, need for ICU hospitalization after surgery) influenced by variables (Gender; Age; Hospital category; Haematology consultation before surgery; Number of years in the practice of anaesthesia and Number of patients with sickle cell disease managed per year), the binary logistic regression model author listed as:

“Z = α + β1Gender + β2Age + β3Hospital Category + β4 Haematology consultation before surgery + β5 Number of years in practice of anaesthesia + β6 Number of sicklers 110 managed per year + μ.”

There are several questions about this model:

  1. Missing variable of Hospital Category in your report table 4.
  2. Author used 10% and 5% two levels of probability in this model, author should list out the detail.
  3. For the entrance of variables of gender and hematology consultant before surgery, did author perform a single factor analysis? Was there significant difference? If not, why choose it?
  4. The sample size is too small, authors entered 6 variables in this model, at least 90 samples should be included to validate your conclusion.

Overall, this manuscript described the differences of anesthesiologist management over sickle cell disease patients, based on a questionnaire survey from 35 anaesthesiologists within Cameroon. The final conclusion didn’t give any lucid results or constructive suggestions. If it is to report the practices of Anesthesiologists in the management of sickle cell disease patient, the sample size is too small to represent the whole.
